# CLASS-INCREMENTAL LEARNING WITH PARAMETER-EFFICIENT CROSS-TASK PROMPTS

## ABSTRACT

Class-Incremental Learning (CIL) aims to learn deep models on sequential tasks continually, where each new task includes a batch of new classes and deep models do not have access to task-ID information at the inference time. Recent vast pre-trained models (PTMs) have achieved outstanding performance by prompt technique in practical CIL without the old samples (rehearsal-free) and with a memory constraint (memory-constrained): Prompt-extending and Prompt-fixed methods. However, prompt-extending methods need a large memory buffer to maintain an ever-expanding prompt pool and meet an extra challenging prompt selection problem. Prompt-fixed methods only learn a fixed number of prompts on one of the incremental tasks and can not handle all the incremental tasks effectively. To achieve a good balance between the memory cost and the performance on all the tasks, we propose a Parameter-Efficient Cross-Task Prompt (PECTP) framework with a prompt retention module (PRM). To make the final learned prompts effective on the whole incremental tasks, PRM constrains the evolution of cross-task prompts' parameters from Outer Prompt Granularity and Inner Prompt Granularity. Extensive experiments show the effectiveness of our method.

## 1 INTRODUCTION

Deep models have achieved outstanding performance when tackling a wide variety of individual machine learning tasks. However, learning deep models on sequential tasks continually (Incremental Learning) remains a formidable challenge (Gomes et al., 2017). Incremental Learning (IL) dynamically learn deep models across different tasks, and often suffers from the performance degradation on previous learned tasks (i.e., *catastrophic forgetting*) (McCloskey & Cohen, 1989). Class-Incremental Learning (CIL) is usually regarded as a challenging setup in IL, where each new task includes a batch of new classes and deep models do not have access to task-ID information at the inference time. Recently, *rehearsal-based* methods can effectively mitigate the forgetting in CIL by keeping few representative samples (*exemplars*) of old tasks in a fixed memory buffer (Rebuffi et al., 2017; Hou et al., 2019). However, these approaches fail in cases with rigorous privacy concerns and severely constrained memory, where the samples of old task are unavailable and the memory buffer is limited. In this paper, we focus on our scope to strategies for the challenging CIL without the exemplars (*rehearsal-free*) and with a memory constraint (*memory-constrained*).

Latest advances in vast pre-trained models (PTMs) have already made a great success in rehearsal-free and memory-constrained CIL, because of PTMs' powerful representation ability. These PTM-based CIL methods bridge the gap between the pre-trained data and sequentially learned tasks' data with parameter-efficient adaptation techniques, e.g., prompt (Jia et al., 2022). With the privacy concern and the memory constraint in practical CIL, the tunable prompts (Wang et al., 2022c) make the frozen pre-trained model capable of adaptive on different tasks effectively and efficiently.

Existing PTM-based CIL methods mainly focus on how to utilize the prompts, and can be briefly separated into two categories: **Prompt-extending**(Wang et al., 2022b;a; Razdaibiedina et al., 2023; Smith et al., 2023) and **Prompt-fixed** methods (Zhou et al., 2023; Yoo et al., 2023). **Prompt-extending** methods need to maintain an ever-expanding prompt pool at the training stage and select suitable prompts from the pool at the inference stage. During training, a novel set of prompts is learned when a new task arrives, with the aim of instructing the PTM to perform conditionally on this current incremental task. After that, these task-specific prompts are stored in a prompt pool that

continually expands as the incremental tasks add sequentially. During inference, a prompt selection strategy is employed to select the suitable prompts for each sample. However, continually expanding the prompt pool can result in the increase of the memory cost, which is not feasible in practical memory-constrained CIL. Besides, the prompt selection strategy not only adds extra computational cost but also encounters a dilemma in modeling the relationship of prompts for different incremental tasks. Another line of work, **Prompt-fixed** methods learn a fixed number of prompts solely on one of the whole incremental tasks (consider the learned task as a *key-task*), and incorporate a feature fusion module to make the key-task prompts generalized on all the tasks. Although Prompt-fixed methods can efficiently save the memory cost, due to the limited knowledge of the single key-task, it is difficult for these fixed number of prompts to instruct PTM to perform well on the remaining incremental tasks. Motivated by the above analysis, in order to achieve an efficient and effective PTM-based method in rehearsal-free and memory-constrained CIL, the key point is to learn a fixed number of effective prompts that can instruct the PTM to perform conditionally on all the tasks.

In this paper, we present a **Parameter-Efficient Cross-task Prompt (PECTP)** framework, a PTM-based approach for practical memory-constrained CIL with a focus on the parameter cost and effectiveness of prompts across different incremental tasks. Our PECTP framework dynamically learns a few number of cross-task prompts, and we propose a **Prompt Retention Module (PRM)** to make these prompts effective on the learned incremental tasks. The PRM module restricts the evolution of cross-task prompts' parameters from two granularity: **Outer Prompt Granularity** (OPG) and **Inner Prompt Granularity** (IPG). Specifically, OPG restricts parameter evolution of prompts by regularizing the output feature of prompt-based PTM. IPG constrains prompt parameter variation by regularizing prompts' parameters themselves. With another classification layer updating scheme about the current task, the final learned cross-task prompts can perform well on the whole incremental tasks. Furthermore, we explore the influence of classification layer updating scheme to the fixed number of cross-task prompts.

To sum up, the main contributions of this paper are: (1) We summarize the prompt-extending and prompt-fixed CIL methods with PTM and propose a PECTP framework for memory-constrained and rehearsal-free CIL, which learns a fixed number of cross-task prompts on the whole incremental tasks. (2) We design a novel PRM to restrict the evolution of cross-task prompts from OPG and IPG, which ingeniously makes the updated cross-task prompts effective on the learned incremental tasks. and (3) Extensive experiments over benchmark datasets demonstrate the effectiveness of our PECTP method in performance and memory cost against the existing prompt-fixed and prompt-extending PTM-based CIL methods.

## 2 RELATED WORK

**Incremental Learning**    Numerous methods have been explored to address catastrophic forgetting (Masana et al., 2022; Mai et al., 2022), and they can be roughly categorized into three main categories: ($i$) architecture-based, ($ii$) rehearsal-based, and ($iii$) regularization-based (Kirkpatrick et al., 2017; Zenke et al., 2017). Architecture-based methods (Rusu et al., 2016; Yoon et al., 2017; Li et al., 2019; Loo et al., 2020; Mallya & Lazebnik, 2018; Serra et al., 2018; Ke et al., 2020) segregate components within the deep model for each task by expanding the network or constraining the learning rate of important parameters towards previous tasks. However, most of these methods require a task ID for inference (Wortsman et al., 2020), which is not suitable for challenging CIL. Rehearsal-based methods (Buzzega et al., 2020; Cha et al., 2021; Rebuffi et al., 2017; Wu et al., 2019; Ebrahimi et al., 2020; Pham et al., 2021; Zhao et al., 2021; De Lange et al., 2021; Van de Ven & Tolias, 2019; Wang et al., 2018) mitigate forgetting by replaying real samples or generated samples of previous tasks. However, these methods are unsuitable in rehearsal-free and memory-constrained CIL. In contrast, our method, PECTP, not only conducts inference without relying on the task ID but also introduces a negligible number of additional parameters.

**Prompt Learning**    Prompt is introduced to enable a small set of trainable parameters to instruct a fixed pre-trained model to perform conditionally (Liu et al., 2023; Li & Liang, 2021; Lester et al., 2021; Huang et al., 2022; Zhou et al., 2022b;a). Prompt has been utilized in a wide range of tasks (incremental learning) with the pre-trained model. Incremental learning with prompt-based PTM typically fall into two main categories: Prompt-fixed and Prompt-extending. Prompt-fixed methods employ a fixed number of prompts for deliberate learning a single incremental task and keep these

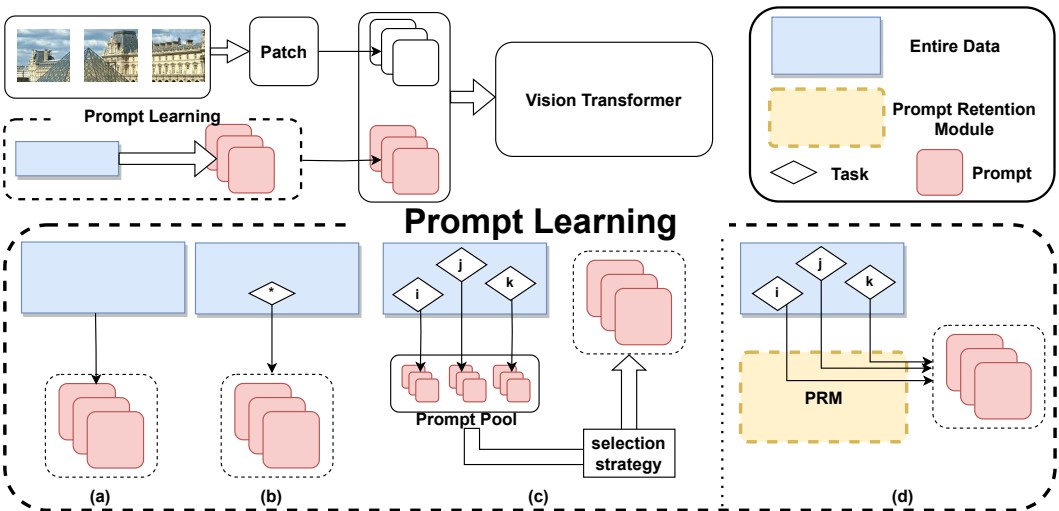

Figure 1: The different piplines of the PTM-based CIL methods. $(a)$ Joint Training: prompts are learned on the entire data, $(b)$ Prompt-fixed methods: prompts are learned on a key-task from all the incremental tasks, $(c)$ Prompt-extending methods: novel sets of prompts are learned on each incremental task, and $(d)$ our PECTP.

prompts fixed throughout subsequent tasks (Zhou et al., 2023). However, these methods often suffer from limited representative capabilities, which results in inadequate guidance for the PTM. In contrast, prompt-extending methods continually learn novel prompts for each novel incremental task, accumulating them in an expanding prompt pool (Smith et al., 2023; Wang et al., 2022c;b;a; Douillard et al., 2022). Nevertheless, continually growing the prompt pool can lead to increased memory cost, making it unsuitable in memory-constrained CIL. Conversely, PECTP effectively instructs the PTM across the all incremental tasks with only a small number of prompts.

## 3 PREREQUISITE

### 3.1 MEMORY-CONSTRAINED AND REHEARSAL-FREE CLASS-INCREMENTAL LEARNING

Formally, Class-Incremental Learning (CIL) aims to learn a deep model on sequential tasks with novel classes. We denote the sequence of tasks as $\left\{\mathcal{T}^1, \mathcal{T}^2, \ldots\right\}$, where $\mathcal{D}^k = \left\{\left(x_i^k, y_i^k\right)\right\}_{i=1}^{n_k}$ is the training data corresponding to task $k$ with $n_k$ training samples. Here, each input sample $x_i^k \in \mathbb{R}^n$ belongs to class $y_i^k \in Y_k$, where $Y_k$ is the label space of task $k$. There are no overlapping classes between tasks (i.e., $Y_k \cap Y_{k'} = \oslash$ if $k \neq k'$). CIL suffers the forgetting problem because the model is trained only on the current task and evaluated over all the learned tasks (all encountered classes are denoted as $\mathcal{Y}_k = Y_1 \cup, \ldots, Y_k$). In *memory-constrained* and *rehearsal-free* CIL, the memory buffer is limited and samples of previous tasks can not be replayed when learning the current task. A deep image classification model is denoted as $\phi_{\theta,w}(x) = g_w(f_\theta(x))$, where $f_\theta(x) : \mathcal{R}^{|\mathcal{D}^k|} \to \mathcal{R}^d$ is a feature extractor with weights $\theta$, and $g_w(\cdot) : \mathcal{R}^d \to \mathcal{R}^{|\mathcal{Y}_k|}$ is a classification layer with weights $w$. After learning the task $k$, the goal is to learn a $\phi_{\theta,w}(\cdot)$ that can performs well on $\mathcal{Y}_k$ with a memory constraint. Recent PTM-based CIL methods (Wang et al., 2022c) usually utilize a pre-trained model (PTM) with powerful representation capability, such as Vision Transformer (ViT), as the initialization for $f_\theta(x)$.

### 3.2 PRE-TRAINED MODEL-BASED CIL WITH PROMPTS

Pre-trained models encounter domain gap problem between pre-trained data and the downstream data. Parameter-efficient adaptation techniques, e.g., prompts, are proposed to address this issue, with the purpose to instruct the PTM to perform conditionally, e.g., Visual Prompt Tuning (VPT) (Jia et al., 2022). Based on VPT, there are two kinds of PTM-based CIL methods with prompts: Prompt-extending and Prompt-fixed CIL methods (shown in Figure 1).

**VPT** Given a frozen pre-trained ViT model $f$ and a set of learnable parameters $\mathcal{P} = \{p_l, l = 1, 2, \ldots\}$, namely prompts. We denote the ViT model with tunable prompts: $f + \mathcal{P}$, as VPT ($\phi$) and $\mathcal{D}$ as the entire data of the downstream task. The objective function is a classification loss on the downstream task and defined as follows:

$$\mathcal{L}_{cls} = \sum_{(x,y) \in \mathcal{D}} \mathcal{L}\left(g_w\left(f\left([x; \mathcal{P}]\right)\right), y\right), \tag{1}$$

where $\mathcal{L}$ is a binary-cross entropy loss, $w$ stands for the parameters of classification layer and $[\cdot; \cdot]$ indicates concatenation on the sequence length dimension.

**Prompt-extending CIL methods** Given the sequential of tasks $\{\mathcal{T}^k, k = 1, 2, \ldots\}$, prompt-extending CIL methods maintain a prompt pool $\mathcal{C} = \{\mathcal{P}^k, k = 1, 2, \ldots\}$ during training. For the task $k$, $\mathcal{P}^k = \{p_l^k, l = 1, 2, \ldots\}$ is the task-specific prompts and deliberately learned on task $k$ by the following loss function:

$$\mathcal{L}_{cls} = \sum_{(x,y) \in \mathcal{D}^k} \mathcal{L}\left(g_{w^k}\left(f\left([x; \mathcal{P}^k]\right)\right), y\right), \tag{2}$$

where $f + \mathcal{P}^k$ is a VPT model $\phi^k$ for $k$-th incremental task and $\mathcal{D}^k$ is the corresponding training data. During inference, a selection strategy $\mathcal{F}(\cdot|x)$ is employed to select the suitable prompts $\mathcal{P}^*$ for each sample $x$:

$$\mathcal{P}^* = \mathcal{F}\left(C|x\right). \tag{3}$$

Prompt-extending CIL methods raise concerns from two perspectives: (1) increasingly expanding the capacity of $\mathcal{C}$ leads to failure in practical memory-constrained CIL, and (2) the design of the prompt selection strategy $\mathcal{F}(\cdot|x)$ has a significant impact on the final performance.

**Prompt-fixed CIL methods** Prompt-fixed CIL methods only learn a fixed number of prompts $\mathcal{P} = \{p_l, l = 1, 2, \ldots\}$ on one of the whole incremental tasks (i.e., key-task $\mathcal{D}^*$) with a classification loss and freeze them in the remaining tasks:

$$\mathcal{L}_{cls} = \sum_{(x,y) \in \mathcal{D}^*} \mathcal{L}\left(g_w\left(f\left([x; \mathcal{P}]\right)\right), y\right). \tag{4}$$

During inference, they incorporate a feature fusion module $\mathcal{G}(f\left(x; \mathcal{P}\right), f\left(x\right))$ to enhance the prompts' generalization ability on all tasks. Nevertheless, the fixed number of prompts learned on the key-task still suffer the degradation in other tasks. It is necessary to develop an efficient and effective PTM-based method in rehearsal-free and memory-constrained CIL.

## 4 PTM-BASED CIL WITH PARAMETER-EFFICIENT CROSS-TASK PROMPT

In this section, we introduce our PECTP method in detail, which can enable a fixed number of prompts to efficiently instruct the PTM to perform effectively on the whole incremental tasks (as shown in Figure 2). Due to the memory-constraint in CIL, PECTP utilizes a fixed number of prompts, instead of adopting a continuously expanding prompt pool. To make these prompts generalized on the whole incremental tasks, PECTP updates the prompts on each incremental task, rather than solely on the key-task. Then, the classification loss on the current task $k$ is defined as follows:

$$\mathcal{L}_{cls} = \sum_{(x,y) \in \mathcal{D}^k} \mathcal{L}\left(g_w\left(f\left([x; \mathcal{P}]\right)\right), y\right). \tag{5}$$

$\mathcal{L}_{cls}$ makes the fixed number of prompts effective on the current task. To make these prompts effective on the previous learned tasks, we propose a prompt retention module (PRM). Our PRM restricts the evolution of cross-task prompts' parameters from IPG (Section 4.1) and OPG (Section 4.2). The training and inference procedure of our PECTP are described in Section 4.3.

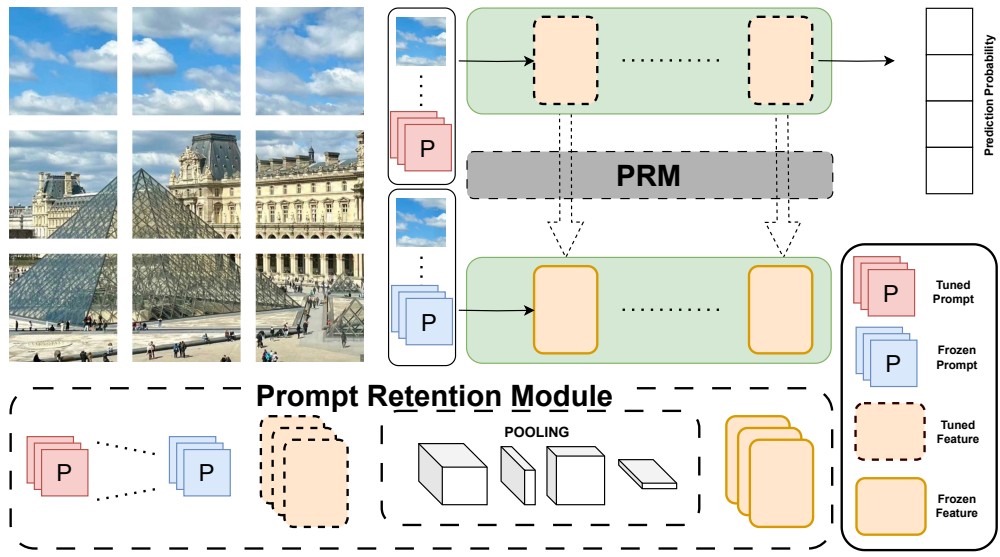

Figure 2: An architecture of the PECTP framework.

## 4.1 PRM FROM INNER PROMPT GRANULARITY

Our PRM constrains prompt parameter variation from Inner Prompt Granularity. Specifically, while learning the $k$-th incremental task, we approximate the prompts' parameters in $\phi^k$ to that in $\phi^{k-1}$. We denote each transformer block in $\phi^k$ as $f_i^k, i = 1, 2, \ldots, N$ and the input feature of the $i$-th transformer block as $d_i^k$. The output of $i$-th transformer block can be formulated as follows:

$$\left[c_{i+1}^k; e_{i+1}^k; \_\right] = f_i^k\left(d_i^k\right), i = 1, 2, ..., N \tag{6}$$

$$\hat{y} = g_w^k(c_{N+1}^k), \tag{7}$$

where $d_i^k = \left[c_i^k; e_i^k; p_i^k\right]$ is the input feature, $c_i^k \in \mathbb{R}^D$ denotes the $[CLS]$, $e_i^k \in \mathbb{R}^{L_g \times D}$ denotes the embedding of the input image with sequence length $L_g$ and embedding dimension $D$, $p_i^k \in \mathbb{R}^{L_p \times D}$ denotes the prompts with prompts' length $L_p$.

Simultaneously, the classification layer $g_w^k(\cdot)$ is used to map the final transformer block's $[CLS]$ embedding, $c_{N+1}^k$, into a predicted class probability distribution $\hat{y}$. When learning the $k$-th incremental task, the prompts in $\phi^k$ should keep the knowledge of the learned prompts in $\phi^{k-1}$ and obtain the knowledge of current task $k$. By stacking over the transformer block axis, the total prompts can be formulated as $\mathbf{P}^k = \left[p_1^k, p_2^k, \ldots, p_N^k\right] \in \mathbb{R}^{N \times L_p \times D}$. In order to make $\mathbf{P}^k$ effective on task $k - 1$, we impose a Inner Prompt Granularity Loss $\mathcal{L}_{\text{IPG}}$ between $\mathbf{P}^k$ and $\mathbf{P}^{k-1}$:

$$\mathcal{L}_{\text{IPG}}\left(\mathbf{P}^{k-1}, \mathbf{P}^k\right) = \sum_{n=1}^{N} \sum_{w=1}^{L_p} \sum_{h=1}^{D} \left\|p_{n,w,h}^{k-1} - p_{n,w,h}^k\right\|^2. \tag{8}$$

## 4.2 PRM FROM OUTER PROMPT GRANULARITY

Our PRM restricts parameter evolution of prompts from Outer Prompt Granularity, which regularizes the output feature of prompt-based PTM. Previous Convolution Neural Network (CNN) based CIL methods employ regularization on the output feature of CNN (Douillard et al., 2022; 2020). Our Outer Prompt Granularity PRM introduces a set of prompt constraints, not only over the final output feature but also over the intermediate output feature of each transformer block.

While learning the $k$-th incremental task, we denote the output feature of each transformer block in the model $\phi^k$ as $\mathbf{h}_i^k = \left[c_{i+1}^t, e_{i+1}^k, \_\right], i = 1, 2, \ldots, N$. Simultaneously, the model $\phi^{k-1}$ can also extract features from each transformer block and the corresponding output features are denoted as $\mathbf{h}_i^{k-1}, i = 1, 2, \ldots, N$. By stacking over the transformer block axis, the total output feature can be formulated as $\mathbf{h}^k = \left[\mathbf{h}_1^k, \mathbf{h}_2^k, ..., \mathbf{h}_N^k\right] \in \mathbb{R}^{N \times (1 + L_g + L_p) \times D}$. Each element of $\mathbf{h}^k$ can be denoted as $h_{n,w,h}^k$, where $n$ represents the block, and $w, h$ stands for patch and dimension axis, respectively.

To approximate the prompts' parameters in $\phi^k$ to that in $\phi^{k-1}$, we aim to make the output features generated by $\phi^k$ similar as the features generated by $\phi^{k-1}$. A simple implementation is to ensure that the output features generated by both models are identical at every feature dimension (point by point). We refer to the corresponding loss as $\mathcal{L}_{\text{OPG-Point-Wise}}$:

$$\mathcal{L}_{\text{OPG-Point-Wise}}\left(\mathbf{h}^{k-1}, \mathbf{h}^k\right) = \sum_{n=1}^{N} \sum_{w=1}^{1+L_g+L_p} \sum_{h=1}^{D} \left\| \mathbf{h}_{n,w,h}^{k-1} - \mathbf{h}_{n,w,h}^k \right\|^2. \tag{9}$$

However, part of features generated by each transformer block are weakly important or even not related to the final prediction (Chen et al., 2023; Rao et al., 2021). $\mathcal{L}_{\text{OPG-Point-Wise}}$ can make $\phi^k$ hard to fetch the truely important features, which results in degradation in the learned tasks. Additionally, extreme constraint can disrupt the flexibility to gain novel knowledge from the current task. To address this issue, we propose a set of soft constraints on the statistic distribution of the original output features $\mathbf{h}^k$. $\mathbf{h}^k$ includes the block, patch, and dimension axis. Then we propose to obtain the distribution knowledge of $\mathbf{h}^k$ from these three axes by average pooling operation. Specifically, (1) pooling over the block axis calculates the output feature distribution from different blocks:

$$\mathcal{L}_{\text{OPG-Block-Wise}}\left(\mathbf{h}^{t-1}, \mathbf{h}^t\right) = \sum_{w=1}^{1+L_g+L_p} \sum_{h=1}^{D} \left\| \sum_{n=1}^{N} \mathbf{h}_{n,w,h}^{t-1} - \sum_{n=1}^{N} \mathbf{h}_{n,w,h}^t \right\|^2, \tag{10}$$

(2) pooling over the patch axis calculates the output feature distribution from different locations:

$$\mathcal{L}_{\text{OPG-Patch-Wise}}\left(\mathbf{h}^{t-1}, \mathbf{h}^t\right) = \sum_{n=1}^{N} \sum_{h=1}^{D} \left\| \sum_{w=1}^{1+L_g+L_p} \mathbf{h}_{n,w,h}^{t-1} - \sum_{w=1}^{1+L_g+L_p} \mathbf{h}_{n,w,h}^t \right\|^2, \tag{11}$$

and (3) pooling over the dimension axis can calculate the output feature distribution from both the blocks and locations:

$$\mathcal{L}_{\text{OPG-Dimension-Wise}}\left(\mathbf{h}^{t-1}, \mathbf{h}^t\right) = \sum_{n=1}^{N} \sum_{w=1}^{1+L_g+L_p} \left\| \sum_{h=1}^{D} \mathbf{h}_{n,w,h}^{t-1} - \sum_{h=1}^{D} \mathbf{h}_{n,w,h}^t \right\|^2. \tag{12}$$

After obtaining the distribution of the original output features $\mathbf{h}^k$, we make the distribution information of $\phi^k$ approximated to that of $\phi^{k-1}$. Such distribution-level constraints can be considered as a form of soft constraints, effectively mitigating hard constraints that prevent the model from learning new knowledge from the current incremental task. With these OPG soft constraints, it is feasible to strike an optimal balance between learning new task knowledge and preserving old task knowledge.

### 4.3 Training and Inference

**Training**  Our model is trained with three parts of losses: (1) the classification loss $\mathcal{L}_{cls}$, a binary-cross entropy to learn on the current incremental task, (2) an Inner Prompt Granularity loss $\mathcal{L}_{\text{IPG}}$ in PRM to regularize the prompts' parameter themselves, and (3) an Outer Prompt Granularity loss $\mathcal{L}_{\text{OPG}}$ in PRM to restrict parameter evolution of prompts by regularizing the output feature of prompt-based PTM. The total loss is:

$$\mathcal{L}_{\{\mathbf{P}, w\}} = \mathcal{L}_{cls} + \alpha \mathcal{L}_{\text{IPG}} + \beta \mathcal{L}_{\text{OPG}}, \tag{13}$$

where $\alpha$ and $\beta$ are two hyperparameters to maintain the balance between learning new task knowledge and preserving old task knowledge.

The performance of $\phi^k$ can be influenced by both classification layer parameters $w$ and the prompts' parameters $\mathbf{P}$ and the initialization of the classification layer can influence the learning process of $\phi^k$ and $\mathbf{P}$. Therefore, we explore the impact of initializing the classification layer during training.

In previous studies, prompt-extending methods typically involve training a new set of prompts and independently $r$einitializing the classification layer for the current incremental task during training. Prompt-fixed methods only train the prompts on a key-task, avoiding the issue of the classification

| Method | CIFAR Inc10 | | CUB Inc10 | | IN-R Inc10 | | IN-A Inc10 | | ObjNet Inc10 | | Omni Inc30 | | VTAB Inc10 | |
|---|---|---|---|---|---|---|---|---|---|---|---|---|---|---|
| | $\bar{A}$ | $A_B$ | $\bar{A}$ | $A_B$ | $\bar{A}$ | $A_B$ | $\bar{A}$ | $A_B$ | $\bar{A}$ | $A_B$ | $\bar{A}$ | $A_B$ | $\bar{A}$ | $A_B$ |
| L2P | 88.34 | 84.57 | 69.69 | 56.01 | 73.82 | 67.13 | 47.16 | 38.48 | 63.78 | 52.19 | 73.36 | 64.69 | 77.11 | 77.10 |
| DualPrompt | 89.69 | 84.14 | 74.84 | 60.84 | 70.32 | 64.80 | 52.56 | 42.68 | 59.27 | 49.33 | 73.92 | 65.52 | 83.36 | 81.23 |
| ADAM-Finetune | 87.12 | 81.23 | 90.98 | **85.58** | 71.29 | 63.35 | 61.57 | 50.76 | 61.41 | 48.34 | 73.02 | 65.03 | **87.47** | 80.44 |
| ADAM-VPT-Shallow | 90.25 | 85.04 | 90.70 | 85.54 | 70.19 | 62.75 | 57.72 | 46.15 | 64.54 | 52.53 | 79.63 | 73.68 | 87.15 | 85.36 |
| ADAM-SSF | 90.61 | 85.14 | 90.67 | 85.37 | 73.07 | 65.00 | 62.81 | 51.48 | 69.15 | 56.64 | 80.53 | 74.00 | 85.66 | 81.92 |
| ADAM-Adapter | 92.24 | 87.49 | 90.96 | 85.11 | 75.08 | 67.20 | 60.53 | 49.57 | 67.18 | 55.24 | 80.75 | 74.37 | 85.95 | 84.35 |
| ADAM-VPT-Deep | 90.40 | 84.62 | 89.48 | 83.42 | 74.46 | 66.47 | 60.59 | 48.72 | 67.83 | 54.65 | 81.05 | 74.47 | 86.59 | 83.06 |
| **PECTP(Ours)** | **92.53** | **87.73** | **91.01** | 85.11 | **77.42** | **70.01** | **66.21** | **55.43** | 70.18 | **58.43** | **81.08** | **74.54** | 87.14 | **86.32** |

Table 1: Average performance and the performance after the last task comparison on seven datasets.

layer continuously expanding. In this paper, we introduce PECTP, which dynamically learns for each new incremental task, enabling the ultimately learned cross-task prompts to be effective for all the tasks. Consequently, the classification layer needs to continuously expand based on the number of the learned tasks. Therefore, we propose to initialize the new classification layer with parameters from a previously learned classification layer, instead of using a simple initialization method (e.g., initializing with 0). Experimental results are in A.4 which demonstrate that our proposed initialization method further keep the prompts' parameters $\mathbf{P}^k$ not far away from $\mathbf{P}^{k-1}$ while learning the current task.

**Inference** Considering that continually training the prompts for each new incremental task in a prompt-based PTM can force the PTM to become more specialized on the incremental tasks, potentially overwriting the general knowledge acquired during pre-training. Therefore, we also utilize a feature fusion module $\mathcal{G}(f(x; \mathcal{P}), f(x))$ to enhance the generalization ability of the cross-task prompts. Additionally, we explore the impact of the feature fusion module and other implementations that could lead to further improvements (A.3).

## 5 EXPERIMENTAL RESULTS

### 5.1 EXPERIMENTAL DETAILS

**Datasets** We follow (Zhou et al., 2023) and conduct the experiments on seven datasets: CIFAR100 (Krizhevsky et al., 2009) (CIFAR), CUB200 (Wah et al., 2011) (CUB), ImageNet-R (Hendrycks et al., 2021a) (IN-R), ImageNet-A (Hendrycks et al., 2021b) (IN-A), ObjectNet (Barbu et al., 2019) (ObjNet), Omnibenchmark (Zhang et al., 2022) (Omni) and VTAB (Zhai et al., 2019). As described in (Zhou et al., 2023), the last four datasets have a large domain gap with the pre-trained dataset ImageNet. ImageNet-A and ObjectNet include the challenging samples that PTMs with ImageNet can merely handle, while Omnibenchmark and VTAB contain diverse classes from multiple complex realms. To construct the CIL setting, 200 classes are sampled from ObjectNet and ImageNet-A (300 classes from Omnibenchmark, 50 classes from VTAB). Different CIL settings are represented as Inc$n$: each CIL setting includes several tasks and each incremental task consists of $n$ new classes.

**Training Details** Our PECTP method is implemented based on the classical prompt-fixed CIL method ADAM-VPT-deep (Zhou et al., 2023). We utilize the PTM ViT-B/16-IN21K, which is pre-trained on ImageNet21K. On each incremental task, we train PECTP with the same hyper-parameters in (Zhou et al., 2023) (e.g., learning rate, epoch, weight decay, the number of prompts). As for the hyper-parameter $\alpha$ and $\beta$ of our PRM module, we utilize 1/3.5e5 and 1/4e2 as the default setting on CIFAR100. More details are included in the supplemental material.

**Evaluation Protocol** We evaluate the performance on all seen classes after learning each new incremental task. $\mathcal{A}_b$ denotes the accuracy after the $b$-th task. We report the performance after the last incremental task ($\mathcal{A}_B$) and the average performance across all tasks ($\bar{\mathcal{A}} = \frac{1}{B} \sum_{b=1}^{B} \mathcal{A}_b$).

### 5.2 COMPARISON TO PREVIOUS METHODS

In this section, we compare our method with recent PTM-based CIL methods: L2P (Wang et al., 2022c), DualPrompt (Wang et al., 2022c), CODA-Prompt (Smith et al., 2023), SimpleCIL (Zhou

| Method | | CIFAR Inc10 (10 tasks) | | IN-R Inc10 (20 tasks) | |
|---|---|---|---|---|---|
| | | $\mathcal{A}_B$ | Prompt Num | $\mathcal{A}_B$ | Prompt Num |
| Prompt-extending | L2P | 83.36 | 50 | 69.33 | 100 |
| | DualPrompt | 81.27 | 205 | 68.64 | 405 |
| | CODA-Prompt | 84.59 | 205 | **73.93** | 405 |
| Prompt-fixed | ADAM-VPT-Deep | 83.26 | 12 | 70.07 | 12 |
| | **PECTP(Ours)** | **86.27** | **12** | 73.20 | **12** |

Table 2: Performance and memory cost (Prompt Num) comparison on CIFAR100 and ImageNet-R.

| Method | CIFAR Inc10 | | IN-A Inc10 | |
|---|---|---|---|---|
| | $\bar{\mathcal{A}}$ | $\mathcal{A}_B$ | $\bar{\mathcal{A}}$ | $\mathcal{A}_B$ |
| Baseline | 90.19 | 84.66 | 62.19 | 50.03 |
| Baseline+OPG | 92.43 | 87.66 | 65.48 | 53.92 |
| Baseline+IPG | 91.70 | 87.60 | 61.41 | 49.11 |
| Baseline+IPG+OPG | **92.59** | **87.94** | **66.21** | **55.43** |

(a)

| Method | CIFAR Inc10 | | IN-A Inc10 | |
|---|---|---|---|---|
| | $\bar{\mathcal{A}}$ | $\mathcal{A}_B$ | $\bar{\mathcal{A}}$ | $\mathcal{A}_B$ |
| Baseline | 90.19 | 84.66 | 62.19 | 50.03 |
| OPG-L | 91.88 | 87.02 | 61.01 | 51.61 |
| OPG-B | 92.43 | 87.69 | 65.23 | 53.98 |
| OPG-P | 90.99 | 86.02 | 63.00 | 51.94 |
| OPG-D | 90.84 | 87.02 | 60.33 | 50.16 |
| OPG-B+D | **92.43** | **87.66** | **65.48** | **53.92** |
| OPG-L+D+P | 90.92 | 85.94 | 60.87 | 51.22 |

(b)

Table 3: Results of ablating components of PRM (i.e., OPG, IPG) and results of ablating variants of OPG (i.e., blcok, patch and dimension) on CIFAR100 and ImageNet-A.

et al., 2023), and ADAM (Zhou et al., 2023). ADAM has different variants with various adaptation techniques (i.e., ADAM-Finetune, ADAM-VPT-Shallow, ADAM-VPT-Deep, ADAM-SSF, ADAM-Adapter). L2P, DualPrompt, and CODA-Prompt are Prompt-extending methods. ADAM-VPT-Deep, ADAM-VPT-shallow and Our proposed PECTP are Prompt-fixed methods.

We show the performance of these methods on seven datasets in Table 1. On typical CIL datasets CIFAR100, CUB and ImageNet-R, our method achieves 92.53%, 91.01% and 77.42% on $\bar{\mathcal{A}}$, outperforming the baseline ADAM-VPT-Deep by 2.13%, 1.53% and 2.96%, respectively. On the challenging ImageNet-A and ObjectNet datasets, our method achieves 66.21% and 70.18% on $\bar{\mathcal{A}}$, surpassing Adam-Adapter by 5.68% and 2.35%, respectively. On Omnibenchmark and VTAB datasets, the $\bar{\mathcal{A}}$ of our method is 7.16% and 3.78% higher than that of DualPrompt respectively.

We also make a detailed comparison about the memory cost between our method and other prompt-based PTM CIL methods. The memory cost of these methods are utilized to keep the PTM and prompts, and the difference of these methods mainly comes from the number of prompts (denoted by Prompt Num). The experimental results datasets are shown in Table 2. On CIFAR100 including 10 tasks, We can observe that our PECTP method achieves highest performance and lowest Prompt Num after learning the last incremental task. On ImageNet-R which consists of 20 tasks, our method has comparable performance than CODA-Prompt while the prompt number of our method is only around 3% of CODA-Prompt. All of the results illustrate that our PECTP method can make a good balance between CIL performance and the memory cost.

## 5.3 ABLATION EXPERIMENTS

**Effect of PRM module** As described Section 5.1, we select the ADAM-VPT-Deep (Zhou et al., 2023) as the Baseline of our PECTP method. We have conducted the experiments to validate the effectiveness of our PRM (OPG and IPG). As shown in Table 3a, both OPG and IPG can improve the performance of Baseline on CIFAR100 and ImageNet-A datasets. While utilizing OPG and IPG simultaneously, our method achieves the highest performance, illustrating that these two granularity are necessary and complementary for keeping the old task knowledge of prompts.

**Different OPG Implementation** Our OPG method can be implemented with different strategies and we have conducted experiments to validate the influence of different OPG implementations. We have several OPG variants: (1) OPG-logit (OPG-L): using a logit-based loss. (2) OPG-block (OPG-B): $\mathcal{L}_{\text{OPG-Block-Wise}}$. (3) OPG-patch (OPG-P): $\mathcal{L}_{\text{OPG-Pacth-Wise}}$. (4) OPG-dimension (OPG-D): $\mathcal{L}_{\text{OPG-Dimension-Wise}}$. The experimental results are shown in Table 3b. The performance of OPG-L is lower than other OPG variants, especially on ImageNet-A, which aligns with the findings in

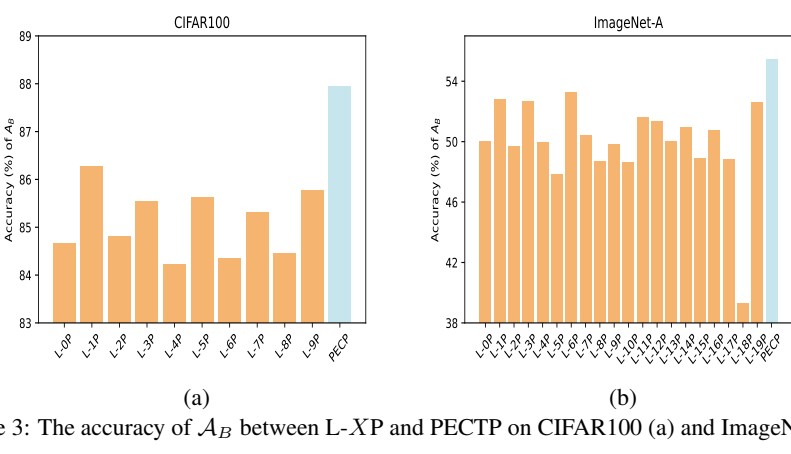

Figure 3: The accuracy of $\mathcal{A}_B$ between L-$X$P and PECTP on CIFAR100 (a) and ImageNet-A (b).

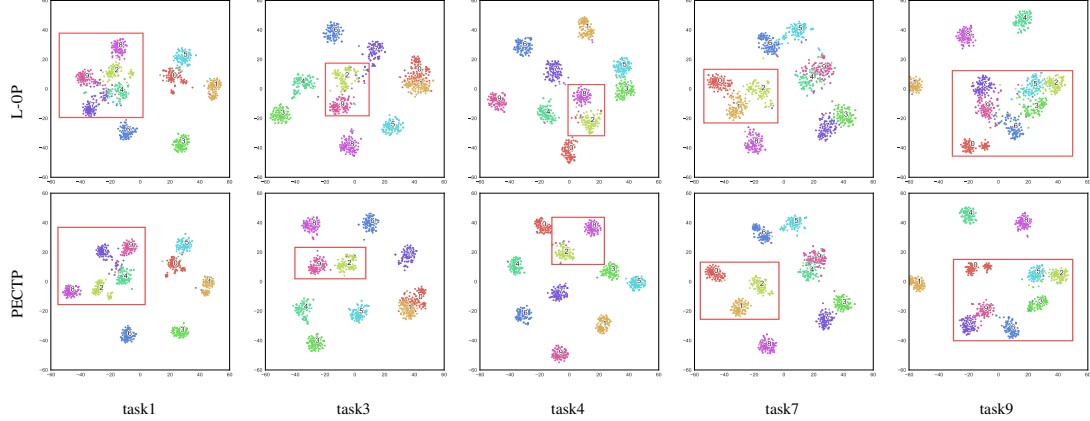

Figure 4: T-SNE visualization of features obtained by L-0P and PECTP on each incremental task in CIFAR100.

the study conducted by (Yang et al., 2022). The performance of OPG-B+D achieve the highest performance, thus our PECTP utilize this variant as the default setting.

**Our Cross-Task Prompt vs. Key-Task Prompt** Our PECTP method utilizes the cross-task prompts and the prompt-fixed methods employ the key-task prompts. To further validate the effectiveness of our method, we conduct the experiments to compare the cross-task prompts and the key-task prompts on CIFAR100 and ImageNet-A. Supposing there are 10 tasks (denoted as $\{D^0, D^1, ..., D^9\}$), we select different task as the key-task to obtain the key-task prompts. For simplicity, the key-task prompts learned on the $D^X$ task are denoted as L-$X$P. The accuracy of our PECTP and different key-task prompts is shown in Figure 3a and 3b. The accuracy of PECTP consistently surpasses that of L-$X$P on both datasets. Additionally, we observe that the selection of the key-task significantly influences prompts' learning. Furthermore, we visualize the extracted features using T-SNE. As shown in Figure 4, features are insufficiently extracted with L-0P, resulting in a fuzzy classification boundary. In contrast, PECTP maintains a clear classification boundary on each incremental task. Further results are in A.2

## 6 CONCLUSION

In this paper, we make a detailed analysis about the prompt-extending and prompt-fixed PTM-based CIL methods, and design a memory-efficient CIL framework with parameter-efficient cross-task prompts. Our PECTP method utilizes a prompt retention module to restrict parameter evolution of cross-task prompts from the outer prompt granularity and inner prompt granularity, which can effectively keep the learned knowledge of the prompts after learning the new incremental task. We perform intensive evaluations of our method and other PTM-based methods with prompts, showing the effectiveness of our method.

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
