# OpenReview forum: "Class-Incremental Learning with Parameter-Efficient Cross-Task Prompts"
_ICLR.cc/2024/Conference — Submitted to ICLR 2024_

### Official Review · Reviewer_9kbr · 2023-10-23

**Soundness:** 2 fair
**Presentation:** 3 good
**Contribution:** 3 good
**Rating:** 5
**Confidence:** 3

**Summary:**

In this work, the author introduces the use of a parameter-efficient cross-task prompt for pre-trained models to tackle the challenges of class incremental learning in a rehearsal-free and memory-constrained manner. Specifically, unlike memory-based incremental learning methods, this approach obviates the need to retrain old samples. In comparison to the previous "Prompt-fixed" and "Prompt-extending" methods, the author proposes a more parameter-efficient manner of extracting relevant prompts.

**Strengths:**

The foundation of this idea appears solid. The parameter efficient way seems memory friendly than the previous methods.

To validate this approach, it was tested across seven popular benchmarks, yielding promising results in some datasets.

**Weaknesses:**

However, I have several concerns:

1. Data Leakage from Pre-trained Model: The model has been pre-trained on ImageNet21k, and subsequently, the author employs a subset of this, ImageNet, for the incremental learning task. This presents a potential data leakage concern, as the pre-trained model has already been exposed to the entirety of ImageNet during its initial training. This practice may inadvertently violate the foundational principles of class incremental learning.

2. Not being directly immersed in prompt learning, I'm concerned that the prompts might inadvertently leak information or task-id specifics to the broader model, especially when compared with prior class incremental learning methodologies, which only obtains the task id based on model itself, no prompts are used.

3. Reproducibility of experiments is crucial. The primary results in Table 1 seem to have been conducted only once, without any mention of variance or average performance.

4. While the number of prompts is detailed in the table, there's a noticeable lack of information regarding model size, computational complexity, and the execution time spent on distinct model components.

5. The paper neglects the discussion of certain challenging tasks. Managing tasks in larger scales, such as when the number of tasks is 50 or 100, can be formidable. I'm curious about the performance of this method under such rigorous testing scenarios.

6. A missing baseline: Given that the pre-trained model might already contain a wealth of information or representations, including those of the dataset under testing, there's a potential risk of the model being over-fitted to the dataset in question. I strongly recommend that the author includes performance metrics without the prompt, thereby delineating the actual gains derived from this approach.

**Questions:**

Regarding the weaknesses, my main concern is data leakage from the pre-trained model itself. If the author can address this concern during the rebuttals, I will increase the scores accordingly.

---

> ### Author Response · Authors · 2023-11-20
> **Response to Reviewer 9kbr**
>
> Thank you for your detailed review of our paper and constructive feedback! We have revised our paper and addressed your concerns. We summarize below the changes we made.
>
>
>
> Q1. "A potential data leakage concern: as the pre-trained model has already been exposed to ImageNet21k during its initial training, which may violate the foundational principles of class incremental learning. Therefore, A missing baseline: I strongly recommend that the author includes performance metrics without the prompt, thereby delineating the actual gains derived from this approach."
>
> A1: *PTM-based CIL methods* do exhibit data leakage in ***Usual CIL benchmarks***, as shown in [4]. However, [4] also introduced **four new benchmarks where Data leakage phenomenon does not exist**. Notably, [4] introduces a novel baseline for *PTM-based CIL methods* called **SimpleCIL**. **SimpleCIL** performs directly on incremental tasks **without any prompting or fine-tuning**. The corresponding results can be found in the **third row** of the tables below.
>
> | ViT-B/16-IN21K   | CIFAR B0Inc10 |           | CUB B0Inc10 |           | IN-R B0Inc10 |           |
> | ---------------- | ------------- | --------- | ----------- | --------- | ------------ | --------- |
> |                  | $A$           | $A_{B}$   | $A$         | $A_B$     | $A$          | $A_B$     |
> | L2P              | 88.34         | 84.57     | 66.69       | 56.01     | 73.82        | 67.13     |
> | DualPrompt       | 89.69         | 84.14     | 74.84       | 60.84     | 70.32        | 64.80     |
> | **SimpleCIL**    | **87.13**     | **81.26** | **90.96**   | **85.16** | **61.99**    | **54.55** |
> | ADAM-Finetune    | 87.12         | 81.23     | 90.98       | 85.58     | 71.29        | 63.35     |
> | ADAM-VPT-Shallow | 90.25         | 85.04     | 90.70       | 85.54     | 70.19        | 62.75     |
> | ADAM-SSF         | 90.61         | 85.14     | 90.67       | 85.37     | 73.07        | 65.00     |
> | ADAM-Adapter     | 92.24         | 87.49     | 90.96       | 85.11     | 75.08        | 67.20     |
> | ADAM-VPT-Deep    | 90.40         | 84.62     | 89.48       | 83.42     | 74.46        | 66.47     |
> | PECTP(Ours)      | 92.53         | 87.73     | 91.01       | 85.11     | 77.42        | 70.01     |
>
>
>
> | ViT-B/16-IN21K   | IN-A B0Inc10 |           | Obj B0Inc10 |           | Omni B0Inc30 |           | VTAB B0Inc10 |           |
> | ---------------- | ------------ | --------- | ----------- | --------- | ------------ | --------- | ------------ | --------- |
> |                  | $A$          | $A_B$     | $A$         | $A_B$     | $A$          | $A_B$     | $A$          | $A_B$     |
> | L2P              | 47.16        | 38.48     | 63.78       | 52.19     | 73.36        | 64.69     | 77.11        | 77.10     |
> | DualPrompt       | 52.56        | 42.68     | 59.27       | 49.33     | 73.92        | 65.52     | 83.36        | 81.23     |
> | **SimpleCIL**    | **60.50**    | **48.44** | **65.45**   | **53.59** | **79.34**    | **73.15** | **85.99**    | **84.38** |
> | ADAM-Finetune    | 61.57        | 50.76     | 61.41       | 48.34     | 73.02        | 65.03     | 87.47        | 80.44     |
> | ADAM-VPT-Shallow | 57.72        | 46.15     | 64.54       | 52.53     | 79.63        | 73.68     | 87.15        | 85.36     |
> | ADAM-SSF         | 62.81        | 51.48     | 69.15       | 56.64     | 80.53        | 74.00     | 85.66        | 81.92     |
> | ADAM-Adapter     | 60.53        | 49.57     | 67.18       | 55.24     | 80.75        | 74.37     | 85.95        | 84.35     |
> | ADAM-VPT-Deep    | 60.59        | 48.72     | 67.83       | 54.65     | 81.05        | 74.47     | 86.59        | 83.06     |
> | PECTP(Ours)      | 66.21        | 55.43     | 70.18       | 58.43     | 81.08        | 74.54     | 87.14        | 86.32     |
>
>
>
> The total results can be found above. Specifically, CIFAR, CUB, and ImageNet-R are widely adopted benchmark datasets for CIL. However, due to the data overlap between ImageNet-based benchmarks and the pre-trained dataset, ImageNet is **unsuitable** for evaluating *PTM-based CIL methods*. Therefore, we follow [4] to evaluate our method on four new benchmarks: ImageNet-A, ObjectNet, OmniBenchmark, and VTAB. These new benchmarks not only have **no overlap with ImageNet** but also **exhibit a large domain gap between tasks**, making it challenging for PTMs to handle.
>
> Notably, despite training on an extremely large-scale dataset, SimpleCIL fails to achieve high accuracy on certain tasks, such as ImageNet-R and ImageNet-A. In contrast, PECTP **consistently outperforms SimpleCIL on seven benchmarks**, especially on those without overlap.

---

> > ### Author Response · Authors · 2023-11-20
> > **(Continued) Response to Reviewer 9kbr**
> >
> > Furthermore, employing PTMs for CIL can indeed lead to data leakage. However, numerous experimental results indicate that directly using pretrained models still falls short of the upper bound on incremental task performance. Therefore, our work, along with a series of prior studies [1-5], focuses on exploring how to fully leverage powerful pretrained knowledge, especially when there is a significant domain gap between incremental tasks and the pretraining task.
> >
> > Finally, introducing pretrained models into CIL poses additional research challenges. Addressing how to continuously learn from incremental tasks, acquire knowledge from new incremental tasks, reduce forgetting of old incremental task knowledge, and avoid overwriting pretrained knowledge while maintaining a balance of attention on downstream tasks are all valuable research questions.
> >
> >
> >
> > Q2. "Not being directly immersed in prompt learning, I'm concerned that the prompts might inadvertently leak information or task-id specifics to the broader model, especially when compared with prior class incremental learning methodologies, which only obtains the task id based on model itself, no prompts are used."
> >
> > A2: Firstly, we need to explain the differences between *PTM-based CIL methods* and *Usual CIL methods*.
> >
> > In *Usual CIL methods*, whether through updating all network parameters dynamically via regularization terms [7] or expanding network structures [6], the number of trainable parameters is substantial compared to *PTM-based CIL methods* (as shown in the table of Q4). Therefore, *usual CIL methods* can leverage the network itself to acquire information about the task, such as the task ID.
> >
> > In *PTM-based CIL methods*, on the other hand, to minimize overwriting to the pretrained knowledge acquired by the PTMs, a relatively small number of trainable parameters (e.g., prompts) is typically introduced to guide the PTM in adapting to downstream learning tasks. Consequently, in PTM-based methods, **specific task-related knowledge (e.g., visual features or task information like the task ID) is encoded into the prompts.**
> >
> > Q3. "Reproducibility of experiments is crucial. The primary results in Table 1 seem to have been conducted only once, without any mention of variance or average performance."
> >
> > A3: We want to clarify that we follow [3] and **run our benchmarks for five different shuffles of the task class order** and report the mean and standard deviation of these runs. We do this with a consistent seed (different for each trials) so that results can be directly compared. Here is a summary of the results:
> >
> >
> >
> > | ViT-B/16-IN21K          | CIFAR Inc10  |              | CUB Inc10    |              | IN-R Inc10   |              | IN-A Inc10   |              |
> > | ----------------------- | ------------ | ------------ | ------------ | ------------ | ------------ | ------------ | ------------ | ------------ |
> > |                         | $A$          | $A_{B}$      | $A$          | $A_{B}$      | $A$          | $A_{B}$      | $A$          | $A_B$        |
> > | L2P                     | 88.14+/-0.32 | 84.33+/-0.41 | 66.60+/-0.30 | 56.01+/-0.16 | 73.52+/-0.65 | 67.20+/-0.41 | 47.56+/-0.93 | 38.88+/-0.39 |
> > | DualPrompt              | 89.61+/-0.22 | 84.02+/-0.29 | 74.80+/-0.17 | 60.81+/-0.11 | 70.12+/-0.96 | 64.92+/-0.56 | 52.76+/-1.01 | 42.38+/-1.30 |
> > | SimpleCIL               | 87.13        | 81.26        | 90.96        | 85.16        | 61.99        | 54.55        | 60.50        | 48.44        |
> > | ADAM-VPT-Shallow        | 89.31+/-0.91 | 84.96+/-0.79 | 90.15+/-0.98 | 85.37+/-0.36 | 70.59+/-1.26 | 62.2+/-1.31  | 58.11+/-0.85 | 47.48+/-1.29 |
> > | ADAM-VPT-Deep(Baseline) | 90.29+/-0.32 | 84.95+/-0.42 | 89.18+/-0.47 | 83.88+/-0.91 | 73.76+/-0.76 | 66.77+/-0.75 | 62.77+/-3.69 | 51.15+/-3.34 |
> > | PECTP(Ours)             | 92.49+/-0.19 | 88.09+/-0.16 | 91.0+/-0.21  | 84.69+/-0.52 | 78.33+/-0.64 | 70.28+/-0.19 | 65.74+/-1.22 | 54.66+/-0.99 |
> >
> >
> >
> > The above results illustrate that in **different shuffles of the task class order across various datasets**, the variance of PECTP proposed in this paper is smaller compared to other baseline methods. This demonstrates the robustness and stability of our method.

---

> > > ### Author Response · Authors · 2023-11-20
> > > **(Continued) Response to Reviewer 9kbr**
> > >
> > > Q4. "While the number of prompts is detailed in the table, there's a noticeable lack of information regarding model size, computational complexity, and the execution time spent on distinct model components."
> > >
> > > A4: Thank you for your suggestion. We would like to provide a detailed explanation of the specific calculation method for the number of prompts in *PTM-based CIL methods*.
> > >
> > > Firstly, in existing *PTM-based CIL methods*, the description of **'prompt' refers to a set of prompts instead of a single prompt.** For example, in L2P, $P_{i} \in \mathbb R^{L_{P}\times D}$, where $L_{P}$ is the number of single prompts, and each $P_{i}$ is stored in a prompt pool. Therefore, for L2P, the total number of prompts is *$L_{P} \times$ the number of prompts* .
> > >
> > > Furthermore, current *PTM-based CIL methods* follow VPT[6] for prompt handling, which has two variants: VPT-Deep and VPT-shallow. For instance, in DualPrompt[2], e-prompts: $e_{i} \in \mathbb R^{L_{e}\times D}$ are inserted into the 3-5 layers of the VIT encoder, while g-prompt: $g_{j} \in \mathbb R^{L_{g}\times D}$ is inserted into the 1-2 layers of the VIT encoder. Therefore, for DualPrompt, the total number of prompts is: *the number of e-prompts $\times$ $L_{e}\times$ inserted layers + the number of g-prompts $\times$ $L_{g}\times$ inserted layers*.
> > >
> > > Here is a summary of the results:
> > >
> > >
> > >
> > > | CIFAR Inc10 | ViT-B/16-IN1K |     $A_B$     |   Overhead    |                          |                |                  |                   |
> > > | ----------- | ------------- | :-----------: | :-----------: | :----------------------: | :------------: | :--------------: | :---------------: |
> > > |             |               |               | Prompt Number | Learnable Parameters (M) |    Flops(G)    | Training Time(s) | Selecting Time(s) |
> > > | extend      | DualPrompt    |  83.05+1.16   |      602      |      3.92 (72.59 X)      | 35.19 (2.03 X) |   26.06+/-0.40   |    1.37+/-0.16    |
> > > |             | CODA          |  86.25+0.74   |     4000      |      2.39 (44.25 X)      | 35.17 (2.03 X) |   26.26+/-0.48   |    1.49+/-0.04    |
> > > | fixed       | L2P++         |  82.50+1.10   |      200      |      0.57 (10.56 X)      | 35.18 (2.03 X) |   26.00+/-0.38   |    1.22+/-0.13    |
> > > |             | ADAM_VPT_Deep | 83.20+/-0.76  |      60       |      0.054 (1.00 X)      | 17.28 (1.00 X) |   15.58+/-0.47   |         0         |
> > > |             | PECTP         | 86.27+/-0.015 |      60       |      0.054 (1.00 X)      | 17.28 (1.00 X) |   22.95+/-0.33   |         0         |
> > >
> > > In the above table:
> > >
> > > *'Training Time'* is calculated for the entire dataset going through one epoch;
> > >
> > > *'Selecting Time'* is calculated for the time needed to select the corresponding prompt (prompt retrieval) for 1000 samples.
> > >
> > >
> > >
> > > The results illustrate that the proposed PECTP demonstrates a significant performance improvement, even with a much **smaller parameters** compared to baseline methods (DualPrompt, L2P, and CODA are 10, 44, and 72 times larger than PECTP, respectively). Furthermore, **Flops and Total training time also highlight the lower computational overhead** of PECTP compared to other methods, further emphasizing the superiority of our approach. Overall, PECTP proposed in this paper exhibits better efficiency in **both computations and parameters**, making it more practical for application in memory constraint scenarios.

---

> > > > ### Author Response · Authors · 2023-11-20
> > > > **(Continued) Response to Reviewer 9kbr**
> > > >
> > > > Q5. "The results in large number of tasks setup would be helpful to validate the effectiveness of their method."
> > > >
> > > > A5: Thank you for your suggestions. Learning **tasks with long sequences** have consistently been a challenging research area in incremental learning. We followed [8] to **partition tasks based on the number of classes**, and evaluated our approach on different configurations of ImageNet-R: 20 tasks (10 classes/task), 40 tasks (5 classes/task), 50 tasks (4 classes/task), and 100 tasks (2 classes/task).
> > > >
> > > >
> > > >
> > > > |         Dataset         | ImageNet-R $A_B$ |       |       |       |
> > > > | :---------------------: | :--------------: | :---: | :---: | :---: |
> > > > |      Task Numbers       |        20        |  40   |  50   |  100  |
> > > > |           L2P           |      65.86       | 59.22 | 57.28 | 35.56 |
> > > > |       DualPrompt        |      67.87       | 55.22 | 58.61 | 39.66 |
> > > > |        SimpleCIL        |      54.55       | 54.55 | 54.55 | 54.55 |
> > > > | ADAM-VPT-Deep(Baseline) |      66.47       | 64.3  | 60.35 | 54.07 |
> > > > |       PECTP(Ours)       |      70.01       | 68.15 | 66.18 | 59.75 |
> > > >
> > > > The specific implementation results **demonstrate the consistent superiority of the PECTP across task** sequences of varying lengths. However, we also observed significant performance degradation for L2P, DualPrompt, and PECTP when learning extremely long task sequences. For instance, when continuously learning 100 tasks, PECTP only achieved an 8% improvement over SimpleCIL[4], despite incurring a substantial increase in computational overhead. We acknowledge this as an area for future exploration and investigation. Once again, thank you for bringing this issue to our attention.
> > > >
> > > >
> > > >
> > > > **Finally, we hope our clarifications can solve the reviewer's concerns. If there are more questions, please let us know. We are happy to discuss.**
> > > >
> > > >
> > > >
> > > > ## Refs:
> > > >
> > > > [1] Learning to Prompt for Continual Learning, 2022 CVPR
> > > >
> > > > [2] DualPrompt: Complementary Prompting for Rehearsal-free Continual Learning, 2022 ECCV
> > > >
> > > > [3] CODA-Prompt: COntinual Decomposed Attention-based Prompting for Rehearsal-Free Continual Learning, 2023 CVPR
> > > >
> > > > [4] Revisiting Class-Incremental Learning with Pre-Trained Models: Generalizability and Adaptivity are All You Need, 2023 Arxiv
> > > >
> > > > [5] S-Prompts Learning with Pre-trained Transformers: An Occam's Razor for Domain Incremental Learning, 2022 NIPS
> > > >
> > > > [6] PackNet: Adding Multiple Tasks to a Single Network by Iterative Pruning, 2018 CVPR
> > > >
> > > > [7] Overcoming Catastrophic Forgetting in Neural Networks, 2017 PNAS
> > > >
> > > > [8] iCaRL: Incremental Classifier and Representation Learning, 2017 CVPR

---

> ### Author Response · Authors · 2023-11-23
> **Thank you!**
>
> Thank you for your diligent review of our paper. With your valuable comments and suggestions, especially for the Questions about paper revision, we have significantly improved the clarity, fluency, and details of our paper. We sincerely appreciate you and your advice.

---

### Official Review · Reviewer_vcUE · 2023-11-01

**Soundness:** 3 good
**Presentation:** 2 fair
**Contribution:** 3 good
**Rating:** 6
**Confidence:** 5

**Summary:**

The authors propose the prompt-fixed CIL method for rehearsal-free CIL setup. Also, they also propose the Prompt Retention Module (PRM) to regularize the updating prompt parameters, and it utilizes two granularity features called Outer Prompt Granularity (OPG) and Inner Prompt Granularity (IPG).

**Strengths:**

**1 (Well-defined problem).** There was a scalable issue by proposing prompt-extending methods such as DualPrompt and Coda-prompt. However, they propose the prompt-fixed method with PRM which is a regularization term for avoiding forgetting, so they solve the scalable issue as the number of tasks increases.

**2 (Intuitive Idea).** Their proposed method is intuitive, and well-organized. If I have to solve this problem, I also think the approach like this paper.

**3 (Enough Experiment).** There are several experimental results to prove the effectiveness of their method. Most of my concerns are solved via their experiments, but some are still remaining. I wrote them in the weakness section.

**4 (Well-written).** It is easily written for understanding.

**Weaknesses:**

**1 (Table 2).** While L2P has the prompt pool, the number of prompts does not depend on the number of tasks. As such, L2P should be prompt-fixed, not prompt-extending. Also, I'm wondering the results on comparison of PECTP and the other baselines with same prompt number (Either You can increase the prompt number in PECTP or you can decrease the prompt number in other baselines is fine).

**2 (Necessity of Inner Prompt Granularity).** According to Table 3, utilizing both IPG and OPG has the good synergy to enhance the performance. To the best of my understanding, the roles of IPG and OPG are to prevent forgetting knowledge from the previous task and pretrained model, respectively. To validate the hypothesis that I mentioned, the additional ablation study is needed (i.e, we can split the test set for each task). For example, in table 3, baseline+IPG do not forget in the previous task compared to baseline+OPG or baseline, and baseline+OPG has the consistent results for all the tasks.

**3 (Stress Test).** This work is noteworthy when the number of tasks is large (e.g, n_task=1000) due to scaling issue. As such, reporting the results in large number of tasks setup would be helpful to validate the effectiveness of their method.

**Questions:**

I replace this part with weakness section.

---

> ### Author Response · Authors · 2023-11-20
> **Response to Reviewer vcUE**
>
> Thank you for the review, comments, and constructive feedback! We are encouraged that you found our idea promising, our method effective and our writing clear. We provide answers to your comments and questions below.
>
>
>
> Q1. "While L2P has the prompt pool, the number of prompts does not depend on the number of tasks. As such, L2P should be prompt-fixed, not prompt-extending. "
>
> A1: We have reviewed the description of L2P [1] in the original paper and found an error in our understanding. We categorize L2P as a *prompt-fix method* and will make the corrections in the latest version.
>
>
>
> Q2. "Results on comparison of PECTP and the other baselines with same prompt number."
>
> A2: We would like to provide a detailed explanation of the specific calculation method for the number of prompts in *PTM-based CIL methods*.
>
> Firstly, in existing *PTM-based CIL methods*, the description of **'prompt' refers to a set of prompts instead of a single prompt.** For example, in L2P, $P_{i} \in \mathbb R^{L_{P}\times D}$, where $L_{P}$ is the number of single prompts, and each $P_{i}$ is stored in a prompt pool. Therefore, for L2P, the total number of prompts is *$L_{P} \times$ the number of prompts* .
>
> Furthermore, current *PTM-based CIL methods* follow VPT[6] for prompt handling, which has two variants: VPT-Deep and VPT-shallow. For instance, in DualPrompt[2], e-prompts: $e_{i} \in \mathbb R^{L_{e}\times D}$ are inserted into the 3-5 layers of the VIT encoder, while g-prompt: $g_{j} \in \mathbb R^{L_{g}\times D}$ is inserted into the 1-2 layers of the VIT encoder. Therefore, for DualPrompt, the total number of prompts is: *the number of e-prompts $\times$ $L_{e}\times$ inserted layers + the number of g-prompts $\times$ $L_{g}\times$ inserted layers.*
>
> Finally, we strictly reimplement the baseline method according to the description in the original paper [1-3]. Additionally, we agree with the reviewer's perspective that the **performance of PTM-based methods depends significantly on the number of trainable parameters**. Therefore, maintaining an equal number of parameters for comparison will provide meaningful results. Considering that **PECTP introduces fewer trainable parameters** compared to the baseline methods, **we have included three additional variants: PECTP-L, PECTP-D, and PECTP-C, each roughly equivalent in the number of trainable parameters to L2P, DualPrompt, and CODAPrompt[3],** respectively. Here is a summary of the results:
>
> | CIFAR  |               |    A_B     |         Overhead         |                          |
> | :----: | :-----------: | :--------: | :----------------------: | :----------------------: |
> |        |               |            | Addtional Prompt numbers | Learnable Parameters (M) |
> | extend |  DualPrompt   | 83.05+1.16 |         **602**          |    **0.57 (10.56 X)**    |
> |        |     CODA      | 86.25+0.74 |         **4000**         |    **3.92 (72.59 X)**    |
> | fixed  |     L2P++     | 82.50+1.10 |         **200**          |    **2.39 (44.25 X)**    |
> |        | ADAM_VPT_Deep |   83.26    |          **60**          |    **0.054 (1.00 X)**    |
> |        |     PECTP     |   86.27    |          **60**          |    **0.054 (1.00 X)**    |
> |        |    PECTP-L    |   87.82    |         **240**          |    **0.22 (4.07 X)**     |
> |        |    PECTP-D    |   88.14    |         **600**          |    **0.54 (10.0 X)**     |
> |        |    PECTP-C    |   88.28    |         **3600**         |    **3.24 (60.0 X)**     |
> |  NONE  |  UPPER bound  |   90.86    |            0             |            0             |
>
> The results illustrate that the proposed PECTP demonstrates **a significant performance improvement, even with a much smaller parameters** compared to baseline methods (DualPrompt, L2P, and CODA are 10, 44, and 72 times larger than PECTP, respectively). Furthermore, as the number of prompts in PECTP increases the performance **continues to improve and approaches the UPPER bound**. This further emphasizes the superiority of our approach.

---

> ### Author Response · Authors · 2023-11-20
> **(Continued) Response to Reviewer vcUE**
>
> Q3. "According to Table 3, utilizing both IPG and OPG has the good synergy to enhance the performance. To the best of my understanding, the roles of IPG and OPG are to prevent forgetting knowledge from the previous task and pretrained model, respectively. "
>
> A3: We agree with the reviewer's perspective that **there exists two catastrophic forgetting issues in PTM-based methods**:
>
> (a) forgetting knowledge of previously learned incremental tasks and,
>
> (b) forgetting knowledge obtained from the pretraining stage of PTM.
>
> However, we would like to clarify that the **PRM** proposed in our paper aims to **minimize the former issue by reducing it at two granularity**, specifically categorized as **OPG and IPG**. As for the latter one, we follows the approach in [4], utilizing 'feature fusion module' to mitigate the impact of continual learning on the knowledge gained during pretraining.
>
> Finally, we identify a mistake where we mistakenly quoted the results of Baseline on IN-A Inc10 from [4]. It has been corrected in the latest version of the paper, and we have included additional results from ablation experiments on CUB and ImageNet-R. Here is a summary of the results:
>
> | Method               | CIFAR Inc10 |           | IN-A Inc10 |           | CUB Inc10 |           | IN-R Inc10 |           |
> | -------------------- | ----------- | --------- | ---------- | --------- | --------- | --------- | ---------- | --------- |
> |                      | $A$        | $A_{B}$   | $A$        | $A_{B}$ | $A$      | $A_{B}$ | $A$        | $A_{B}$ |
> | SimpleCIL            | 87.13       | 81.26     | 60.50      | 49.44     | 90.96     | 85.16     | 61.99      | 54.55     |
> | Baseline             | 90.19       | 84.66     | 60.59      | 48.72     | 89.48     | 83.42     | 74.46      | 66.47     |
> | Baseline+OPG         | 92.43       | 87.66     | 65.48      | 53.92     | 90.01     | 85.09     | 77.03      | 69.38     |
> | Baseline+IPG         | 91.70       | 87.60     | 61.41      | 49.11     | 89.69     | 84.01     | 75.91      | 67.43     |
> | **Baseline+OPG+IPG** | **92.59**   | **87.73** | **66.21**  | **55.43** | **91.01** | **85.11** | **77.42**  | **70.01** |
>
> The experimental results demonstrate that IPG exerts a stronger constraint on the prompt itself. When the domain gap between incremental learning tasks is small, such as in CIFAR and ImageNet-R, it leads to a greater improvement in performance. Additionally, the results in the last row indicate that OPG and IPG are complementary. **Using them together can significantly reduce the forgetting of knowledge acquired from previously learned incremental tasks**.
>
>
>
> Q4. "The results in large number of tasks setup would be helpful to validate the effectiveness of their method."
>
> A4:  Thank you for your suggestions. Learning tasks with long sequences have consistently been a challenging research area in CIL. We followed [5] to partition tasks based on the number of classes, and **evaluated our approach on different configurations of ImageNet-R: 20 tasks (10 classes/task), 40 tasks (5 classes/task), 50 tasks (4 classes/task), and 100 tasks (2 classes/task).**
>
>
>
> |         Dataset         | ImageNet-R $A_{B}$ |       |       |       |
> | :---------------------: | :----------------: | :---: | :---: | :---: |
> |      Task Numbers       |         20         |  40   |  50   |  100  |
> |           L2P           |       65.86        | 59.22 | 57.28 | 35.56 |
> |       DualPrompt        |       67.87        | 55.22 | 58.61 | 39.66 |
> |        SimpleCIL        |       54.55        | 54.55 | 54.55 | 54.55 |
> | ADAM-VPT-Deep(Baseline) |       66.47        | 64.3  | 60.35 | 54.07 |
> |       PECTP(Ours)       |       70.01        | 68.15 | 66.18 | 59.75 |
>
>
>
> The specific implementation results **demonstrate the consistent superiority of the PECTP across task** sequences of varying lengths. However, we also observed significant performance degradation for L2P, DualPrompt, and PECTP when learning extremely long task sequences. For instance, when continuously learning 100 tasks, PECTP only achieved an 8% improvement over SimpleCIL[4], despite incurring a substantial increase in computational overhead. We acknowledge this as an area for future exploration and investigation. Once again, thank you for bringing this issue to our attention.
>
>
>  If you have any further questions, we are willing to clarify any misunderstandings and address your inquiries to the best of our ability.
>
> ### Refs:
>
> [1] Learning to Prompt for Continual Learning, 2022 CVPR
>
> [2] DualPrompt: Complementary Prompting for Rehearsal-free Continual Learning, 2022 ECCV
>
> [3] CODA-Prompt: COntinual Decomposed Attention-based Prompting for Rehearsal-Free Continual Learning, 2023 CVPR
>
> [4] Revisiting Class-Incremental Learning with Pre-Trained Models: Generalizability and Adaptivity are All You Need, 2023 Arxiv
>
> [5] iCaRL: Incremental Classifier and Representation Learning, 2017 CVPR
>
> [6] Visual Prompt Tuning, 2022 ECCV

---

> ### Author Response · Authors · 2023-11-23
> **Follow up**
>
> We wanted to follow up and gently ask if we are able to address your concerns in our response. As the discussion period is nearing an end, we would appreciate any updates or further questions you may have, and if not, we hope you might consider raising your score. We thank you for your time in advance!

---

### Official Review · Reviewer_2DHh · 2023-11-04

**Soundness:** 2 fair
**Presentation:** 2 fair
**Contribution:** 2 fair
**Rating:** 3
**Confidence:** 5

**Summary:**

The paper presents a prompt-fixed continual learning method based on pre-trained transformer models. By introducing a regularization module for the task prompts, the method constrains the update of prompts. Extensive experiments on benchmark datasets demonstrates the effectiveness of the proposed method.

**Strengths:**

- The paper is easy to understand.
- Using prompting-based method for continual learning is interesting.

**Weaknesses:**

- The idea of adding regularization terms to a fixed set of prompts is not quite persuasive. I have a series of related questions:
1. What is the intuition behind limiting the update of prompts?
2. How is it different from usual regularization-based CL methods, besides the fact that you are doing on prompt parameters instead of all model parameters?
3. Wouldn't limiting the update also limits the ability of learning new tasks?
4. Why using a fixed set of prompts is even better than extending the set of prompts? For example, if the domain gaps are large between different tasks, the fixed set of prompts might not be able to represent them well without conflicts.
- To me it seems the "feature fusion" module is the most interesting part that fixed prompts method work well. However, this is not the main contribution of the method, and the authors did not describe them in detail in the main paper.

**Questions:**

Please see weaknesses.

---

> ### Author Response · Authors · 2023-11-20
> **Response to Reviewer 2DHh**
>
> Thank you for the review, comments, and constructive feedback! We are encouraged that you found our method effective and our writing clear. We provide answers to your comments and questions below.
>
>
>
> Q1. "What is the intuition behind limiting the update of prompts?"
>
> A1: The intuition behind limiting the update of prompts is to **maintain knowledge related to previous tasks within the prompts**. In existing *PTM-based CIL methods* [1-5], task-specific knowledge is encoded into prompts. The task-specific knowledge is associated with each incremental tasks, such as visual features or task information like the task ID, and is updated continually for each incremental task. Due to considerations of storage and computational overhead, we adopt *prompt-fix methods* that use only a fixed set of prompts.
>
> However, if prompts are **continually trained** on each incremental task using cross-entropy loss **without any constraints**, prompts are prone to losing knowledge of previous tasks. For instance, training a set of prompts $\mathcal P$ on task 1 results in $\mathcal P_{1}$, which acquires knowledge about task 1. This implies that utilizing $\mathcal P_{1}$ effectively guides the PTM in classifying the first task. However, when we continue training $\mathcal P_{1}$ on task 2, resulting in $\mathcal P_{2}$, the parameters of $\mathcal P_{2}$ have changed compared to $\mathcal P_{1}$. This means that $\mathcal P_{2}$ cannot effectively guide the PTM in correctly classifying task 1. Therefore, without adding additional constraints during the training，$\mathcal P_{i}$ is prone to losing information about previous tasks: $1,2,...,i-1$.
>
> To verify this point, we provide an additional set of experimental results, where ***PlainCIL* represents continuously learning new task knowledge without adding any constraints to prompts**. *SimpleCIL* [4] represents directly using a PTM to address incremental learning tasks without any prompting or finetuning. Here is a summary of the results:
>
>
>
> | ViT-B/16-IN21K          | CIFAR Inc10      |                  | CUB Inc10       |                  | IN-R Inc10       |                  | IN-A Inc10       |                  |
> | ----------------------- | ---------------- | ---------------- | --------------- | ---------------- | ---------------- | ---------------- | ---------------- | ---------------- |
> |                         | $A$              | $A_B$            | $A$             | $A_B$            | $A$              | $A_B$            | $A$              | $A_B$            |
> | L2P                     | 88.14+/-0.32     | 84.33+/-0.41     | 66.60+/-0.30    | 56.01+/-0.16     | 73.52+/-0.65     | 67.20+/-0.41     | 47.56+/-0.93     | 38.88+/-0.39     |
> | DualPrompt              | 89.61+/-0.22     | 84.02+/-0.29     | 74.80+/-0.17    | 60.81+/-0.11     | 70.12+/-0.96     | 64.92+/-0.56     | 52.76+/-1.01     | 42.38+/-1.30     |
> | SimpleCIL               | 87.13            | 81.26            | 90.96           | 85.16            | 61.99            | 54.55            | 60.50            | 48.44            |
> | **PlainCIL**            | **90.68**        | **87.0**         | **84.30**       | **77.27** | **68.40**        | **61.28**        | **59.95**        |   **49.31**                |
> | ADAM-VPT-Shallow        | 89.31+/-0.91     | 84.96+/-0.79     | 90.15+/-0.98    | 85.37+/-0.36     | 70.59+/-1.26     | 62.2+/-1.31      | 58.11+/-0.85     | 47.48+/-1.29     |
> | ADAM-VPT-Deep(Baseline) | 90.29+/-0.32     | 84.95+/-0.42     | 89.18+/-0.47    | 83.88+/-0.91     | 73.76+/-0.76     | 66.77+/-0.75     | 62.77+/-3.69     | 51.15+/-3.34     |
> | **PECTP(Ours)**         | **92.49+/-0.19** | **88.09+/-0.16** | **91.0+/-0.21** | **84.69+/-0.52** | **78.33+/-0.64** | **70.28+/-0.19** | **65.74+/-1.22** | **54.66+/-0.99** |
>
>
>
>
>
> Q2. "Wouldn't limiting the update also limits the ability of learning new tasks?"
>
> A2: We also concur with the reviewer's perspective that constraining the update of prompts may impede learning for new tasks, consequently limiting the acquisition of new knowledge. However, even in *usual regularization-based CIL methods* [7], there exists a trade-off between new and old knowledge. **Specifically, in usual classic incremental learning approaches, there exists weakening the forgetting of old knowledge by constraining learning for new tasks.**
>
> In summary, we are actively seeking a **more reasonable balance between new and old knowledge**. Additionally, although the motivation behind the proposed method is to efficiently utilize prompt parameters to adapt to highly memory-constrained scenarios, our experimental results indicate that even for some methods with parameters far exceeding PECTP, the proposed method remains comparable. For specific results, please refer to **https://openreview.net/forum?id=4lqo5Jwfnq&noteId=Iv3lvYIpt9**.

---

> > ### Author Response · Authors · 2023-11-20
> > **(Continued) Response to Reviewer 2DHh**
> >
> > Q3: "Why using a fixed set of prompts is even better than extending the set of prompts? For example, if the domain gaps are large between different tasks, the fixed set of prompts might not be able to represent them well without conflicts."
> >
> > A3: *Prompt extending methods* struggle to achieve satisfactory performance due to challenges during **prompt retrieval**, especially when each incremental tasks are highly similar or when there is a small amount of data corresponding to each individual task.
> >
> > **Prompt retrieval** [1-4] refers to the process of prompt selection in PTM-based methods. The selection strategy often depends on methods such as *Attention* [1, 2, 3] or *K-means* [5]. Previous experimental results have indicated **that replacing Task or Domain Identification accuracy with selection strategies usually results in low accuracy [5]**. Additionally, when facing long task sequences, adding a new set of prompts for each task may lead to high similarity between prompts. Given that the total dataset size is fixed, longer task sequences imply a lower training data size for each incremental task. As a result, prompts trained in this manner may have limited representative capacity for task features, thereby exacerbating the challenge during prompt retrieval.
> >
> > In contrast, PECTP not only possesses the advantage of being parameter-efficient but also, by consistently maintaining a fixed set of prompts, **avoids issues associated with prompt retrieval such as 'low accuracy' and 'additional computational overhead'.**
> >
> > In the table below, we evaluated PECTP on different configurations of ImageNet-R: 20 tasks (10 classes/task), 40 tasks (5 classes/task), 50 tasks (4 classes/task), and 100 tasks (2 classes/task). The specific implementation results are presented below, **demonstrating the consistent superiority of the proposed PECTP across task sequences of varying lengths.**
> >
> >
> >
> > |         Dataset         | ImageNet-R $A_B$ |       |       |       |
> > | :---------------------: | :--------------: | :---: | :---: | :---: |
> > |      Task Numbers       |        20        |  40   |  50   |  100  |
> > |           L2P           |      65.86       | 59.22 | 57.28 | 35.56 |
> > |       DualPrompt        |      67.87       | 55.22 | 58.61 | 39.66 |
> > |        SimpleCIL        |      54.55       | 54.55 | 54.55 | 54.55 |
> > | ADAM-VPT-Deep(Baseline) |      66.47       | 64.3  | 60.35 | 54.07 |
> > |       PECTP(Ours)       |      70.01       | 68.15 | 66.18 | 59.75 |
> >
> >
> >
> > Q4. How is it different from usual regularization-based CL methods, besides the fact that you are doing on prompt parameters instead of all model parameters?
> >
> > A4: The differences between PECTP and *usual regularization-based CIL methods* are as follows:
> >
> > (a) In *usual regularization-based CL methods*, the model parameters are **all trainable**, whereas in *PTM-based CIL methods*, parameters are typically **loaded from pre-trained models and fixed**, with only a minimal proportion of prompt parameters being trainable.
> >
> > (b) The trade-off between new and old knowledge in *usual CIL methods* is translated into the updates of prompt parameters in *PTM-based CIL methods*. Therefore, it is necessary to **impose constraints on the learning process of prompts' parameters** to balance the incorporation of new and old knowledge.
> >
> > To the best of our knowledge, we are the first to introduce regularization in *PTM-based CIL methods*, **constraining the updates of prompt parameters at two granularity levels to achieve a balance between new and old knowledge**. Additionally, the proposed PECTP, by utilizing only a fixed set of prompts with a limited length, is memory-friendly in practical CIL scenarios.

---

> > > ### Author Response · Authors · 2023-11-20
> > > **(Continued) Response to Reviewer 2DHh**
> > >
> > > Q5. To me it seems the "feature fusion" module is the most interesting part that fixed prompts method work well. However, this is not the main contribution of the method, and the authors did not describe them in detail in the main paper.
> > >
> > > A5: We would like to clarify that during training, PECTP utilizes the PRM to constrain prompt updates, and during the prediction phase, it incorporates 'feature fusion module' to merge features [4] [9] [10]. **The primary contribution to the excellent performance of PECTP is the proposed PRM **, which constrains prompt updates at two granularity, **rather than the 'feature fusion module'.**
> > >
> > > To verify this point, we provide an additional set of experimental results:
> > >
> > > (a) *Baseline*: **SimpleCIL**[4], directly conducts evaluations on incremental tasks using PTM without prompting or fine-tuning;
> > >
> > > (b) *V-PECTP*: Vanilla-PECTP, **employs PRM to limit prompts' updates but does not utilize the feature fusion module during inference**. Compared to the Baseline, it only adds a trainable set of prompts.
> > >
> > > (c) *Baseline + Baseline*: During inference, 'feature fusion module' is applied to **merge the features obtained from two SimpleCIL models**. The parameters are equivalent to PECTP.
> > >
> > > (d) *Baseline + V-PECTP*: PECTP, **employs PRM** to restrict prompts' updates, and 'feature fusion module' is used to combine the features from both *V-PECTP* and SimpleCIL during inference. The parameters are equivalent to *Baseline + Baseline*.
> > >
> > > | Method              | CIFAR |       | ImageNet-A |       |
> > > | ------------------- | ----- | ----- | ---------- | ----- |
> > > |                     | $A$   | $A_B$ | $A$        | $A_B$ |
> > > | *Baseline*          | 87.13 | 81.26 | 60.50      | 49.44 |
> > > | *V-PECTP*           | 91.91 | 87.04 | 63.48      | 50.36 |
> > > | *Baseline+Baseline* | 88.33 | 82.13 | 60.79      | 49.50 |
> > > | *Baseline+V-PECTP*  | 92.53 | 87.73 | 66.21      | 55.43 |
> > >
> > > The results above illustrate that, compared to the *Baseline*, although *Baseline+Baseline* doubles the parameters, the performance improvement is marginal. On the other hand, *V-PECTP* significantly outperforms both of the aforementioned methods, further indicating that **the performance improvement in our proposed PECTP is not reliant on an increase in model parameters but rather achieved through PRM to effectively control the balance between new and old knowledge.**
> > >
> > > Additionally, we also observe that in *PTM-based CIL methods*, two issues of catastrophic forgetting are prevalent:
> > >
> > > (a) Forgetting knowledge of previously learned incremental tasks, and
> > >
> > > (b) Forgetting knowledge obtained from the pretraining stage of PTM.
> > >
> > > In this paper, our proposed **PRM aims to minimize the former issue by addressing it at two granularity, OPG and IPG**. As for the latter issue, we adopt the approach in [4], **using feature fusion to mitigate the impact of continual learning on the knowledge gained during pretraining**.
> > >
> > > Furthermore, although 'feature fusion module' is not the main contribution of our paper, we have **explored different fusion methods**, such as 'point-wise-add': *PointAdd* and 'concatenation': *concat*, as detailed in Appendix A.3.
> > >
> > > | Fusion Method | CIFAR |       | ImageNet-A |       |
> > > | ------------- | ----- | ----- | ---------- | ----- |
> > > |               | $A$   | $A_B$ | $A$        | $A_B$ |
> > > | PointAdd      | 92.59 | 87.94 | 66.21      | 55.43 |
> > > | Concat        | 92.4  | 87.86 | 66.29      | 55.63 |
> > >
> > > Finally, these experimental results and analyses will be incorporated into the latest version of the paper to further enhance its readability.
> > >
> > >
> > >
> > > **Finally, if you have any further questions, we are willing to clarify any misunderstandings and address your inquiries to the best of our ability.**
> > >
> > >
> > >
> > > #### Refs:
> > >
> > > [1] Learning to Prompt for Continual Learning, 2022 CVPR
> > >
> > > [2] DualPrompt: Complementary Prompting for Rehearsal-free Continual Learning, 2022 ECCV
> > >
> > > [3] CODA-Prompt: COntinual Decomposed Attention-based Prompting for Rehearsal-Free Continual Learning, 2023 CVPR
> > >
> > > [4] Revisiting Class-Incremental Learning with Pre-Trained Models: Generalizability and Adaptivity are All You Need, 2023 Arxiv
> > >
> > > [5] S-Prompts Learning with Pre-trained Transformers: An Occam's Razor for Domain Incremental Learning, 2022 NIPS
> > >
> > > [6] PackNet: Adding Multiple Tasks to a Single Network by Iterative Pruning, 2018 CVPR
> > >
> > > [7] Overcoming Catastrophic Forgetting in Neural Networks, 2017 PNAS
> > >
> > > [8] iCaRL: Incremental Classifier and Representation Learning, 2017 CVPR
> > >
> > > [9] DER: Dynamically Expandable Representation for Class Incremental Learning, 2021 CVPR
> > >
> > > [10] Class-Incremental Learning with Strong Pre-trained Models, 2022 CVPR

---

> ### Author Response · Authors · 2023-11-23
> **Follow up**
>
> Dear Reviewer 2DHh,
>
> We thank the reviewer for their engagement in the rebuttal process. As the discussion period is nearing an end, we would appreciate any updates or further questions you may have, and if not, we hope you might consider raising your score. We thank you for your time in advance!
>
> Best,
>
> Paper 2478 Authors

---

### Author Response · Authors · 2023-11-23
**Invitation to reply**

Dear Reviewers,

As we are approaching the close of the discussion period, we would like to make one last effort in engaging with the reviewers. To reiterate, we have incorporated the reviewer feedback and revised our paper. We believe that we have adequately addressed all the raised concerns.



In light of these changes and clarifications, we ask that the reviewers, especially those who had a negative initial impression of our paper, to reevaluate their stance. We understand the burden of reviewing, however the success of the peer-review system hinges on a active dialogue between the involved parties, and we sincerely hope for your engagement.

Best,

Paper 2478 Authors

---

### Meta-Review · Area_Chair_1GbQ · 2023-12-11

**Metareview:**

In this paper, the authors present a prompt-fixed continual learning method based on pre-trained transformer models. This paper received mixed reviews: two reviewers were negative towards this paper, while the third one tended to accept it.

After carefully checking the paper, reviews, and authors' feedback, I tend to reject this paper. This paper is less polished, and the quality is relatively low compared to the ICLR standard. I suggest the authors update the paper according to reviewers' comments and submit it to the next venue.

**Justification For Why Not Higher Score:**

The overall quality of this paper is not satisfactory. For example, the experiments are conducted on small-scale datasets, and many recent class-incremental learning baselines are not compared. The authors did a good job in the rebuttal. However, this is not enough to make the current version accepted.

**Justification For Why Not Lower Score:**

N/A

---

### Decision · Program_Chairs · 2024-01-16

Reject